# QUOTA CONSTRAINTS FOR
# DIVERSITY INTERVENTIONS IN SUBSET SELECTION

**Neeraja Abhyankar**
neeraja.abhyankar@gmail.com

## ABSTRACT

The combinatorial optimization problem of subset selection is often modeled as maximizing a set function that captures inter-element dependencies under some capacity/matroid constraints. In this paper, we examine this problem under "quota constraints" where the selected subset must meet some minimum group-wise quotas. We provide algorithms for two popular scenarios extended to the quota-constrained setting and make an empirical case for their applicability to fair subset selection.

## 1 INTRODUCTION

The subset selection problem is found in various domains: hiring, choosing a cohort of candidates from a pool of people, recommending a small set of news articles, summarizing data, building a financial portfolio, etc. where one tries to choose an often capacity-constrained set $S \subset V$ (the universal set) so as to maximize a utility function $u(S)$. Very often, this utility is also strongly correlated with how *representative* the chosen subset is, of the data at large. As is usually done (Celis et al., 2016; Hesabi et al., 2015), we will break down the *diversity* of a set S into two notions.

**Diversity that is captured in the utility function** $u$**.** e.g. when the objective rewards geometric diversity over some feature space, or when $u$ is submodular (Krause & Golovin, 2014; Lin & Bilmes, 2011; Chen et al., 2018; Feige & Izsak, 2013), i.e. it captures the diminished marginal gain in utility from adding a member to $S$ when another similar member is already present in $S$.

**Diversity that is enforced as an intervention.** This includes constraints imposed on the optimization problem in order to overcome biases in the data/objective (Mitchell et al., 2020; O'neil, 2017; Mehrabi et al., 2021), affirmative action to induce longer-term changes in the macro environment (Kleinberg & Raghavan, 2018; Celis et al., 2020; 2021; Hu & Chen, 2018), or diversification as a means to safeguard against uncertainty (e.g. in a financial portfolio). This notion of diversity is often uncorrelated or inversely correlated with our understanding of utility $u$.

Fairness constraints based on group memberships (Binns, 2020; Dwork et al., 2012) typically enforce group-wise capacity constraints on the chosen subset (e.g. require $S$ to be an independent set of a partition matroid). In practice, however, when wanting representation across intersectional underrepresented groups or a large number of groups, a natural constraint is to enforce a minimum number (hereforth termed as **quotas**) of items chosen from each group. A common example of this is affirmative action policies like Baswana et al. (2019) or variations of the Rooney Rule.

The contribution of this paper is an analysis of quota constraints for two popular and illustrative circumstances: maximizing a monotone submodular function and sampling from a determinantal point process. We provide algorithms for both problems and conclude with an empirical discussion of the utilitarian price of such constraints.

## 2 SUBMODULAR MAXIMIZATION SUBJECT TO QUOTAS

Formally stated, let the elements of a ground set $V$ belong to $p$ protected groups $\{V_j \subseteq V\}_{j=1}^p$ *that may be disjoint or intersecting* with quotas $\{k_j \in \mathbb{R}\}_{j=1}^p$. We wish to choose

$$S = \underset{S \in \mathcal{Q}}{\arg\max}\, u(S) \quad \text{where } \mathcal{Q} = \{A : |A \cap V_j| \geq k_j \ \forall j \in 1, ..., p \text{ and } |A| \leq k\} \qquad (1)$$

where $u$ is a normalized (i.e. $u(\emptyset) = 0$), non-decreasing, submodular function (i.e. $u(a|A) \leq u(a|B) \; \forall a \in V, B \subseteq A \subseteq V$ where $u(a|A)$ stands for $u(A \cup a) - u(A)$).

Note that the set $\mathcal{Q}$ of allowable subsets of $V$ is *not* a matroid. A matroid is characterized as a set of *independent* sets $\mathcal{I} \subseteq 2^U$ that is downward closed, i.e., $A \in \mathcal{I}$ and $B \subseteq A \Rightarrow B \in \mathcal{I}$, and such that all maximal elements of $\mathcal{I}$ have the same cardinality. The theory and applications of maximizing set functions (especially submodular or submodular-supermodular) subject to (one or many) matroidal constraints are extremely well-studied under various circumstances (Nemhauser et al., 1978; Edmonds, 1968; Calinescu et al., 2011; Do & Neumann, 2020; Buchbinder et al., 2019), and the regular greedy heuristic has been shown to have guarantees under several of these (Bai & Bilmes, 2018; Friedrich et al., 2019).

We wish to propose a modification of the regular greedy heuristic for solving (1). In the QUOTA-GREEDY heuristic (Algorithm 1, formally presented in Appendix A), we iteratively build $S$ in two stages – first by restricting our search over members of $V_j$ for whom the quotas $k_j$ have not been satisfied, until such subsets $V_j$ exist, followed by regular greedy addition of elements. For the mere purpose of illustration, we have also outlined CAPACITYGREEDY (Algorithm 2 in Appendix A), where every constraint of the type $|A \cap V_j| \geq k_j$ is converted to $|A \cap V \setminus V_j| \leq k - k_j$. While this allows us to frame the problem as maximization over an intersection of matroids, note that the problem that Algorithm 2 solves is **not** equivalent to (1).

While asserting an approximation guarantee for Algorithm 1 is relegated for future work, we provide a discussion in Appendix B to compare QUOTAGREEDY for (1) with the regular greedy heuristic under a partition matroid constraint, i.e. when the groups $\{V_j\}_{j=1}^p$ are disjoint and exhaustive.

## 3 SAMPLING FROM A QUOTA CONSTRAINED DPP

Let us now extend the concept from Celis et al. (2018) to the quota-constrained setting, in order to understand the probability space of subsets satisfying geometric diversity (with more diverse sets being more likely to be sampled) as well as group-wise diversity (satisfied for every sample). Instead of restricting group memberships to follow exact capacity constraints, we define the problem as: sample $S$ according to the distribution

$$P(S) \propto \begin{cases} \det(X_S X_S^\top) & \text{if } S \in \mathcal{Q} = \{A : |A \cap V_j| \geq k_j \; \forall j \in 1, ..., p \text{ and } |A| \leq k\} \\ 0 & \text{otherwise} \end{cases} \quad (2)$$

where $X \in \mathbb{R}^{|V| \times m}$ is a feature matrix characterizing the ground set. We can slightly modify their linear-time Sample-And-Project algorithm to extend to the Quota-constrained case (see Appendix C) and still expect a very analogous performance guarantee to hold under the conditions specified in the original paper. Moreover, since a quota constraint is less rigid than an exact capacity constraint, we also expect the *price of fairness* as defined in Celis et al. (2018) – the KL-divergence $D_{\mathrm{KL}}(q\|r)$ (where $q$ is the distribution defined by (2) and $r$ is the unconstrained distribution over $\{A : |A| \leq k\}$) to be upper bounded by the exact capacity counterpart.

## 4 DISCUSSION

The motivation to analyze quota constraints arose from the observation that often, members belonging to underrepresented groups contribute disproportionately more toward the utility of the cohort as a whole when chosen alongside more members of the same group, as opposed to chosen in isolation (e.g. "token diversity hires" or samples in summaries which lack context). An attempt is made to demonstrate that quota-like constraints can be employed as an intervention, alongside tweaks to existing algorithms without incurring an extra *price of fairness*, i.e. without compromising on the attained utility of the subset as a whole, as compared to group capacity constraints. Along with a possible approximation guarantee, another conjectured property of QUOTAGREEDY not captured in the current formalism is that filling up underrepresented group quotas with high-leverage members first (as opposed to when the gains to $u$ are diminished after reaching capacity on the overrepresented groups) will result in higher quality representation across said groups. In Appendix D, we try out CAPACITYGREEDY and QUOTAGREEDY on realistically constructed synthetic data and observe that the latter results in much better group-wise diversity while incurring a negligible hit to utility.

ACKNOWLEDGEMENTS

I would like to express sincere gratitude to Prof. Jamie Morgenstern of the University of Washington, for providing insights, pointers, and having discussions during the time this work was done.

URM STATEMENT

The author of this paper is not affiliated with a funded organization or team whose primary goal is research. In addition, the author meets the URM criteria of the ICLR 2023 Tiny Papers Track.

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

## A    APPENDIX: THE QUOTAGREEDY ALGORITHM

---

**Algorithm 1:** QUOTAGREEDY

---

**Input:**  Universal set $V$,
  membership matrix $\mu \in \{0,1\}^{|V| \times p}$ over p groups,
  group quotas $\{k_j\}_{j=1}^p$, total set capacity $k$,
  oracle access to the marginal utility gains $u(a|A) \; \forall a \in V, A \subseteq V$

1 Initialize $S \leftarrow \emptyset$, $u(S) \leftarrow u(\emptyset)$;
  // **Quota-filling stage**
2 $\mathcal{R} \leftarrow \left\{ j : \sum_{v \in S} \mu_{vj} < k_j \right\}$
3 **while** $|\mathcal{R}| > 0$ **do**
      // **Search over all inadequately-represented groups** $\mathcal{R}$
4    $\quad S \leftarrow S \cup \underset{\substack{v \in V \setminus S \\ \sum_{j \in \mathcal{R}} \mu_{vj} > 0}}{\arg\max} \; u(v|S)$
5    $\quad \mathcal{R} \leftarrow \left\{ j : \sum_{v \in S} \mu_{vj} < k_j \right\}$
6 **end**
  // **Regular greedy stage**
7 **for** $t = |S| + 1, ..., k$, **do**
      // **Add the best elements**
8    $\quad S \leftarrow S \cup \underset{v \in V \setminus S}{\arg\max} \; u(v|S)$
9 **end**

---

**Algorithm 2:** CAPACITYGREEDY

---

**Input:**  Universal set $V$,
  membership matrix $\mu \in \{0,1\}^{|V| \times p}$ over p groups,
  group quotas $\{k_j\}_{j=1}^p$, total set capacity $k$,
  oracle access to the marginal utility gains $u(a|A) \; \forall a \in V, A \subseteq V$

1 Initialize $S \leftarrow \emptyset$, $u(S) \leftarrow u(\emptyset)$;
  // **Having a minimum quota for a certain group is equivalent to**
  // **having a maximum capacity for elements not belonging to that group**
2 Set $h_j = k - k_j \; \forall j = 1, ..., p$;
3 $\mathcal{R} \leftarrow \left\{ j : \sum_{v \in S} (1 - \mu_{vj}) == h_j \right\}$
4 **for** $t = 1, ..., k$, **do**
      // **Search over all elements which are**
      // **NOT members of groups** $\mathcal{R}$ **that have reached capacity**
5    $\quad S \leftarrow S \cup \underset{\substack{v \in V \setminus S \\ \sum_{j \in \mathcal{R}} (1 - \mu_{vj}) == 0}}{\arg\max} \; u(v|S)$
6    $\quad \mathcal{R} \leftarrow \left\{ j : \sum_{v \in S} (1 - \mu_{vj}) == h_j \right\}$
7 **end**

---

## B    APPENDIX: TOWARD A GUARANTEE FOR QUOTAGREEDY FOR SUBMODULAR FUNCTION MAXIMIZATION

To potentially aid us in our endeavor, we will use as a starting point, any appropriate result for the partition matroid version of our problem statement. Here, unlike (1), $\{V_j\}_{j=1}^p$ are necessarily disjoint and each of them is to contain exactly $k_j$ elements.

$$S = \underset{S \in \mathcal{M}}{\arg\max}\, u(S) \quad \text{where } \mathcal{M} = \{A : |A \cap V_j| = k_j \; \forall j \in 1, ..., p\} \tag{3}$$

**Lemma B.1** (**Approximation Guarantee for Greedy with a Partition Matroid Constraint**). *If $S_{\mathcal{M}}^{\dagger}$ is the optimal value of (3) and $\widetilde{S}_{\mathcal{M}}$ is the solution of the greedy heuristic on (3), then $u(\widetilde{S}_{\mathcal{M}}) \geq \alpha \, u(S_{\mathcal{M}}^{\dagger})$.*

Nemhauser et al. (1978) have shown $\alpha$ to be $1/2$ and Calinescu et al. (2011) $\left(1 - \frac{1}{e}\right)$ with a randomized method, but we shall continue to denote the approximation guarantee as a variable $\alpha$ so as to allow substituting extensions to the various special cases (e.g. curvature bounds, different oracle models, etc.).

Note that when $\sum_{j=1}^{p} k_j < k$, i.e. when Problem (1) has a feasible solution, and when $u$ is normalized and non-decreasing, any feasible solution of (3) is also a feasible solution of (1).

### B.1 DISCUSSION: APPROXIMATION GUARANTEE FOR ALGORITHM 1

If $S^\dagger$ is the true optimum of Problem (1) for a submodular, non-negative, normalized utility $u$ and for disjoint (but not necessarily exhaustive) subsets $V_j$, and $\widetilde{S}$ is the output of Algorithm 1, then we would like to have an approximation guarantee of the form $u(\widetilde{S}) \geq \gamma\, u(S^\dagger)$ for some $\gamma$.

Below, we will outline some steps in order to develop intuition for how $u(\widetilde{S})$ and $u(S^\dagger)$ can behave.

1. We can always arbitrarily divide $S^\dagger$ into two sets $S_q^\dagger$ and $S_r^\dagger$ such that $S_q^\dagger$ satisfies all constraints in (1) with equality (and hence also all constraints in (3)).

$$S^\dagger = S_q^\dagger \sqcup S_r^\dagger \tag{4}$$

2. Observe that when the groups $V_j$ are disjoint and exhaustive, the "quota-filling stage" in Algorithm 1 is equivalent to the regular greedy heuristic under a partition matroid constraint. Thus,

$$u(\widetilde{S}_{\mathcal{M}}) \geq \alpha u(S_{\mathcal{M}}^\dagger) \tag{5}$$

$$\geq \alpha u(S_q^\dagger) \quad \text{(from the optimality of } u(S_{\mathcal{M}}^\dagger)) \tag{6}$$

3. Let's denote $\widetilde{S} \setminus \widetilde{S}_{\mathcal{M}}$ by $\widetilde{S}_r$.

$$u(\widetilde{S}) = u(\widetilde{S}_r \cup \widetilde{S}_{\mathcal{M}}) \tag{7}$$

$$= u(\widetilde{S}_{\mathcal{M}}) + u(\widetilde{S}_r | \widetilde{S}_{\mathcal{M}}) \tag{8}$$

4. Since we have chosen elements greedily post the quota-filling stage,

$$u(\widetilde{S}) - u(\widetilde{S}_{\mathcal{M}}) = u(\widetilde{S}_r | \widetilde{S}_{\mathcal{M}}) \tag{9}$$

$$\geq \left(1 - \frac{1}{e}\right) \max_{S : |S| \leq k} u(S | \widetilde{S}_{\mathcal{M}}) \tag{10}$$

$$\geq \left(1 - \frac{1}{e}\right) u(S_r^\dagger | \widetilde{S}_{\mathcal{M}}) \tag{11}$$

5. Attempting to bound $u(S^\dagger)$...

$$u(S^\dagger) \leq u(S_q^\dagger) + u(S_r^\dagger) \quad \text{(submodularity)} \tag{12}$$

$$\leq u(S_q^\dagger) + u(S_r^\dagger \cup \widetilde{S}_{\mathcal{M}}) \quad \text{(monotonicity)} \tag{13}$$

$$\leq \frac{1}{\alpha} u(\widetilde{S}_{\mathcal{M}}) + u(S_r^\dagger \cup \widetilde{S}_{\mathcal{M}}) \quad \text{(by 6)} \tag{14}$$

$$\leq \frac{1}{\alpha} u(\widetilde{S}_{\mathcal{M}}) + \frac{e}{e-1} u(\widetilde{S}) - \frac{1}{e-1} u(\widetilde{S}_{\mathcal{M}}) \quad \text{(by 11)} \tag{15}$$

6. Thus,

$$u(\widetilde{S}) \geq \left(1 - \frac{1}{e}\right) u(S^\dagger) + \left[\frac{1}{e-1} - \frac{1}{\alpha}\right] u(\widetilde{S}_{\mathcal{M}}) \tag{16}$$

$$\geq \left(1 - \frac{1}{e}\right) u(S^\dagger) + \left[\textcolor{orange}{\frac{\alpha}{e-1}} - 1\right] u(S_{\mathcal{M}}^\dagger) \tag{17}$$

Since $\alpha < 1$, the orange term is negative, preventing us from ariving at a general guarantee. However, this expression may serve as a starting point to prove a bound in certain special cases.

## C APPENDIX: QUOTA-AWARE-SAMPLE-AND-PROJECT

---

**Algorithm 3:** QUOTASAMPLEANDPROJECT

**Input:** Universal set $V$,
  membership matrix $\mu \in \{0,1\}^{|V| \times p}$ over p groups,
  group quotas $\{k_j\}_{j=1}^p$, total set capacity $k$,
  feature matrix $X \in \mathbb{R}^{|V| \times m}$

1   Initialize $w_v = X_v \in \mathbb{R}^m \ \forall v \in V$
      `// Quota-filling stage`
2   $\mathcal{R} \leftarrow \left\{ j : \sum_{v \in S} \mu_{vj} < k_j \right\}$
3   **while** $|\mathcal{R}| > 0$ **do**
        `// Sample from members belonging to`
        `// inadequately-represented partitions`
4     Sample $\widetilde{v}$ from distribution $\left\{ \frac{||w_{\widetilde{v}}||^2}{\sum_{v \in R} ||w_v||^2} \right\}_{\substack{\widetilde{v} \in V \setminus S \\ \sum_{j \in \mathcal{R}} \mu_{\widetilde{v}j} > 0}}$
5     $S \leftarrow S \cup \widetilde{v}$
        `// Project all feature vectors onto`
        `// the subspace orthogonal to` $\widetilde{v}$
6     Set $w_v \leftarrow \pi_{\widetilde{v}}(w_v) \ \forall v \in V \setminus S$
        `// Recompute inadequately-represented partitions`
7     $\mathcal{R} \leftarrow \left\{ j : \sum_{v \in S} \mu_{vj} < k_j \right\}$
8   **end**
      `// Regular sample-and-project stage`
9   **for** $t = |S| + 1, ..., k$, **do**
        `// Sample from all remaining elements`
10    Sample $\widetilde{v}$ from distribution $\left\{ \frac{||w_{\widetilde{v}}||^2}{\sum_{v \in R} ||w_v||^2} \right\}_{\widetilde{v} \in V \setminus S}$
11    $S \leftarrow S \cup \widetilde{v}$
12    Set $w_v \leftarrow \pi_{\widetilde{v}}(w_v) \ \forall v \in V \setminus S$
13   **end**

---

## D APPENDIX: EXPERIMENTS ON SYNTHETIC DATA

In order to get a feel for how QUOTAGREEDY would play out in practice, we simulate it on some synthetic data. We used a ground set of size $|V| = 100$, selection budget of $k = 20$, and a simple mixture of concave-over-modular utilities $u(A) = \sum_{i=1}^{m} w_i \sqrt{\sum_{a \in A} X_{ai}}$ where $X \in \mathbb{R}_+^{|V| \times m}$ is a randomly drawn feature matrix, $w_i$ are randomly instantiated weights, and $m = 80$.

Observe how $u(A)$ increases monotonically with each feature $X_{ai}$ from the vector $X_a$ if $a \in A$. Hence, to simulate $p = 3$ groups that are underprivileged w.r.t. their value assessed by $u$, we randomly assigned members to groups with some probabilities, and shrunk some of their features by a factor $0 < \beta_p \leq 1$ by setting $X_{ai} \leftarrow \beta_p X_{ai}$ for all $a \in V_p$ and $i \in$ a randomly chosen subset of features.

In the simulation results to follow, we take representative examples of a few configurations of $\{V_j\}_{i=j}^p$ (group membership distributions in light brown), and select subsets using the following 4 methods:
(final membership distributions plotted as histograms; objective values $u(S)$ in plot subtitles)

1. **[red]** A random subset selection

2. **[teal]** Greedy maximization without any group-wise constraints (i.e. over $\{A : |A| \leq k\}$)

3. **[yellow]** CAPACITYGREEDY 2 maximization with quota constraints expressed as capacity constraints (i.e. over $\{A : |A \cap (V \setminus V_j)| \leq (k - k_j) \ \forall j \in 1, ..., p \text{ and } |A| \leq k\}$)

4. **[orange]** QUOTAGREEDY 1 with group-wise quota constraints (over $\{A : |A \cap V_j| \geq k_j \ \forall j \in 1, ..., p \text{ and } |A| \leq k\}$)

## D.1 DISJOINT GROUPS: 1

In this instance, we chose population distributions over 3 **disjoint** groups $\sim \{0.4, 0.2, 0.4\}$ with feature disadvantage vectors $||\beta_2|| > ||\beta_1|| > ||\beta_0|| = 0 \in (0, 1]^m$.
Quota constraints were set as $k_1 = k_2 = k_3 = 4$.

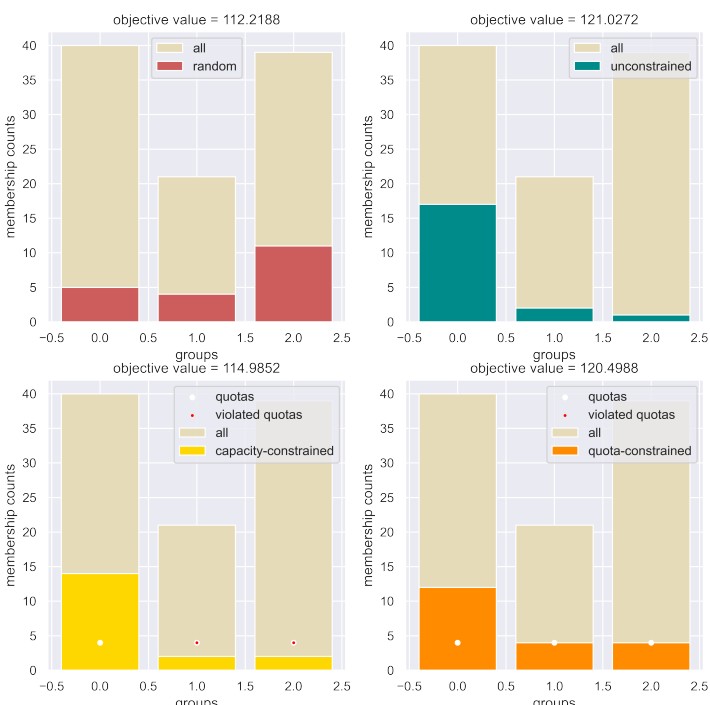

We see an example of the fact that phrasing the problem as in Algorithm 2 and prioritizing the filling of high-gain elements without hitting the quota-sensitive groups may land us in a situation where the set capacity is maxed out (it is not possible to further add elements), we are still satisfying $|A \cap (V \setminus V_j)| \leq (k - k_j) \; \forall j$, but not satisfying $|A \cap V_j| \geq k_j \; \forall i$. The objective attained by QUOTAGREEDY is almost as good as that attained without quota constraints, and much better than a random selection, pointing to a low price of fairness.

## D.2 DISJOINT GROUPS: 2

This instance is identical to the one above, with a reduction in the magnitudes of feature disadvantage vectors $||\beta_p||$.

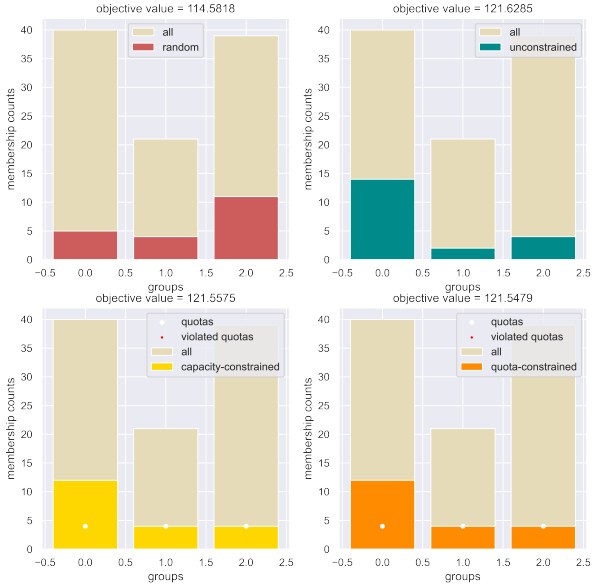

Here, CAPACITYGREEDY outperforms QUOTAGREEDY by a tiny bit, but the difference is still negligible given that both are near the unconstrained max.

### D.3 INTERSECTING GROUPS

In this instance, we sample group assignments over 3 **intersecting** groups with independent probabilities $\sim \{0.4, 0.3, 0.6\}$ with feature disadvantage vectors $||\beta_2|| > ||\beta_1|| > ||\beta_0|| \in (0, 1]^m$. Quota constraints were set as $k_1 = k_2 = k_3 = 4$.

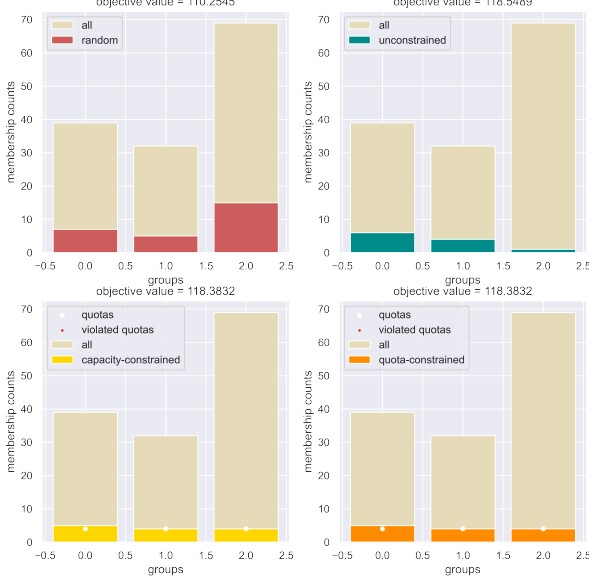

Here too, both algorithms with constraints perform much better than random, and almost as well as group-agnostic greedy.

In the intersecting group paradigm, it is also not rare to see CAPACITYGREEDY and QUOTAGREEDY render different distributions over groups – where the choice of more intersectional members in QUOTAGREEDY result in a distribution with more underrepresented members overall.

**All code for this repository lives here.**

