# OpenReview forum: "Quota Constraints for Diversity Interventions in Subset Selection"
_ICLR.cc/2023/TinyPapers — Submitted to Tiny Papers @ ICLR 2023_

### Official Review · Reviewer_b3dM · 2023-03-24

**Confidence:** 2

**Summary Of Contributions:**

The paper analyzes quota constraints for maximizing monotone submodular function and sampling from a determinantal point process and proposed QuotaGreedy and CapacityGreedy algorithms. The proposed algorithms improve group-wise diversity while not damaging the utility of the constructed synthetic data.

**Rating:**

High Potential (HP): a submission which meets the reviewing criteria and has potential to make an impact on the field

**Strengths And Weaknesses:**

Strengths:
1. The paper is thorough, clear and very well-written.
2. The proposed algorithms seem promising.

Weaknesses:
I don't see any, but I am not very familiar with the studied field.

**Suggested Changes:**

Not really, as I don't research in this field.

---

> ### Author Response · Authors · 2023-06-01
> **Thank you for your review!**
>
> Thanks for reading, reviewing, and rating my paper.

---

### Official Review · Reviewer_7G9w · 2023-04-04

**Confidence:** 3

**Summary Of Contributions:**

The authors in this work examines the problem of quota-constraint in subset selection problem. They analyse this for two popular scenarios: Submodular Maximisation and Sampling from Detrimental Point Process

**Rating:**

Clear, Correct, and Reproducible (CCR): a submission which meets the reviewing criteria

**Strengths And Weaknesses:**

.Strengths:
* The paper clearly introduces the problem and clearly specifies what they want to solve in this paper.

Weakness:
* The major portion of section 2 including algorithm and ablations is available in Appendix, this breaks the flow for the reader.

**Suggested Changes:**

* Try to  minimise introduction and bring algorithm to the main paper rather than in the Appendix

---

> ### Author Response · Authors · 2023-06-01
> **Thank you for your review and feedback!**
>
> Thanks for reading, reviewing, and providing valuable feedback regarding the structure of the paper.
> I have incorporated the gist of the algorithm in the main paper so as not to require the reader to break their flow. However, due to a lack of space, I had to leave the thorough and detailed version of the algorithm in the Appendix. I intend to put up a version on ArXiv where I will re-arrange the sections to make it an easier read.

---

### Meta-Review · Area_Chair_XM4z · 2023-04-05

**Recommendation:** Invite to archive
**Confidence:** 4

**Metareview:**

Pros:
- The authors study an interesting problem.
- The proposed algorithms for ensuring "diversity/quota constraints" make sense.

Cons:
- The theoretical results are not very clear (See detailed feedback).
- Code for experiments is not provided.

**Summary:**

Algorithms for submodular maximization under diversity constraints are considered

**Comments And Feedback To The Authors:**

Should be addressed:
- The theoretical claim is too informally stated in the main body.
- The last two equations (15-->16,16-->17) in the derivation of the main claim of the paper (on page 6) are not clear.
- Some experiments/conclusions could be placed in the main body.
- CAPACITYGREEDY is mentioned without defining in section "Discussion"; in general proposed algorithms need more discussion in the main body.

Writing suggestions:
- replace "1" by "(1)" in the paragraph following equation (2).

**Reason For Not Giving A Higher Recommendation:**

The theoretical results are not very clear, and no code is provided for the experiments.

**Reason For Not Giving A Lower Recommendation:**

N/A

---

> ### Author Response · Authors · 2023-06-01
> **Grateful for your meticulous evaluation!**
>
> Thank you for your review, and extra thanks for finding an issue in a proof in the paper!
>
> After thoroughly revisiting the paper, I have not been able to come up with a fix for the aforementioned problem so far. Consequently, I have removed the claim regarding the approximation guarantee for Algorithm 1 from the paper.
> However, I intend to keep trying to arrive at an algorithm + a guarantee (possibly for a special case or two) and will submit a revised version to ArXiv once I am able to.
>
> Despite this, the rest of the paper remains intact, and I believe that the core idea examined — the introduction of a paradigm of constraints that are non-matroidal, so as to serve as a diversity intervention for a biased objective — will still be of use to the community and will lay the groundwork for future investigations.
>
> ---
>
> As for the remaining concerns you raised, I have addressed them below and incorporated the necessary changes in the camera-ready version.
>
> > Code for experiments is not provided.
>
> As stated at the end of the Appendix Section (D), this was done so as to preserve anonymity. All code lives at https://github.com/neerajaabhyankar/fair-cohort-selection, (link added to the paper as well).
>
> > The theoretical claim is too informally stated in the main body.
> > The last two equations (15-->16,16-->17) in the derivation of the main claim of the paper (on page 6) are not clear.
>
> I again sincerely appreciate you thoroughly reading the proof and finding this out!
> I have replaced the proof with a discussion of how I have approached the problem / a direction toward a proof. For ease of reading, I have also made a slight change to the notation, apologies for any inconvenience caused due to it.
>
> > Some experiments/conclusions could be placed in the main body. CAPACITYGREEDY is mentioned without defining in section "Discussion"; in general proposed algorithms need more discussion in the main body.
>
> I have included the gist of both of the algorithms in the main paper so as not to require the reader to break their flow. However, due to a lack of space, I had to leave the thorough and detailed versions of the algorithms in the Appendix.
>
> ---
>
> Your feedback has been pivotal in improving the clarity of my work, and I’m truly grateful for your meticulous evaluation and thoughtful suggestions.

---

### Decision · Program_Chairs · 2023-04-07

Invite to archive